# Associations between Changes in Body Weight Status and High Blood Pressure among Lithuanian Children and Adolescents during the COVID-19 Pandemic: A Retrospective Cohort Study

**DOI:** 10.3390/nu16193256

**Published:** 2024-09-26

**Authors:** Ieva Stankute, Virginija Dulskiene, Renata Kuciene

**Affiliations:** Institute of Cardiology, Medical Academy, Lithuanian University of Health Sciences, Sukileliu 15, LT-50162 Kaunas, Lithuania; virginija.dulskiene@lsmu.lt (V.D.); renata.kuciene@lsmu.lt (R.K.)

**Keywords:** body mass index, waist circumference, overweight, obesity, abdominal obesity, high blood pressure, children, adolescents

## Abstract

(1) Background: High blood pressure (HBP), overweight, and obesity are common, growing public health problems worldwide. The aim of this study was to evaluate associations between changes in body weight status and HBP among Lithuanian children and adolescents during the COVID-19 pandemic. (2) Methods: In this study, we analysed data on blood pressure and anthropometric measurements of 2430 children and adolescents aged 8–18 years, who participated in both the baseline study conducted before the COVID-19 pandemic (from November 2019 to March 2020) and the follow-up study during the COVID-19 pandemic (from November 2021 to April 2022). Multivariate logistic regression analysis was used to estimate the associations between changes in weight status categories and HBP. (3) Results: At baseline, 17.1% of the subjects had overweight, 5.9% had obesity, 5.6% had abdominal obesity, and 23.7% had HBP, whereas at the follow-up, these percentages increased to 20.1%, 8.2%, 6.8%, and 27.4%, respectively. Compared to schoolchildren who maintained normal weight from baseline to the follow-up period, subjects who newly developed overweight/obesity and those who remained with persistent overweight/obesity had increased odds of HBP, with adjusted odds ratios (aORs) of 1.95 (*p* < 0.001) and 2.58 (*p* < 0.001), respectively. In subjects who transitioned from overweight/obesity to normal weight, the odds of HBP were slightly increased, with an aOR of 1.14 (*p* = 0.598), but the change was not statistically significant (*p* > 0.05). (4) Conclusions: This study observed an increase in the prevalence of overweight, obesity, and HBP among schoolchildren during the COVID-19 pandemic. The study also suggested that changes from normal body weight status at baseline to overweight/obesity during follow-up, especially persistent overweight/obesity, were associated with higher odds of HBP in Lithuanian children and adolescents during the COVID-19 pandemic.

## 1. Introduction

According to an analysis conducted by the Non-Communicable Diseases (NCD) Risk Factor Collaboration (NCD-RisC), in 2022, 94.2 million boys and 65.1 million girls aged 5–19 years had obesity [1]. Obesity deserves acknowledgment as a serious condition that significantly reduces the quality of life for children and, due to its numerous complications, could even lead to disability [2]. The increasing frequency of childhood obesity is a matter of grave concern due to its demonstrated association with a notable rise in the risk of premature death during early adulthood [3]. Research indicates a strong correlation between overweight and obesity in childhood and the development of hypertension later in life [4]. The presence of cardiometabolic risk factors during adolescence can persist into adulthood, amplifying the likelihood of encountering severe complications and mortality [5]. The findings of the longitudinal studies suggested that children who transitioned from overweight or obesity to normal weight during their school age years may have a decreased risk of developing hypertension [6] or HBP [7] in adolescence. This underscores the importance of addressing childhood obesity as a preventive measure against hypertension and its associated health complications in adulthood.

The Government of the Republic of Lithuania enforced the COVID-19 quarantine starting from 16 March 2020 and extended its duration until 1 July 2021, encompassing various stages of restrictions throughout this period. During this time frame, the government implemented a range of measures aimed at controlling the spread of the pandemic, including lockdowns, travel limitations, and social distancing protocols. These measures were adjusted periodically in response to the evolving situation and the guidance from health experts. The overarching goal was to safeguard public health while mitigating the impact of the COVID-19 outbreak on the country’s populace. The Government of the Republic of Lithuania ended the national emergency related to COVID-19 on 1 May 2022 [8]. During lockdown in Lithuania, educational institutions enforced strict measures to curb the spread of COVID-19, necessitating a shift to remote learning. This transition enabled schoolchildren and students to continue their education from home, following the education programs remotely. Such measures aimed to maintain educational continuity while prioritizing public health and safety [9].

A systematic review and meta-analysis by Chang et al. demonstrated significant increases in body weight and body mass index (BMI), and also an increased prevalence of obesity and overweight in children and adolescents during the COVID-19 lockdown period [10]. A systematic review by Karatzi et al. emphasized the deterioration of nutrition and lifestyle among children and adolescents due to increased consumption of unhealthy foods high in sugar, sodium, and fat, increased sedentary and screen time, and decreased physical activity, all of which contributed to increased body weight and abdominal fat accumulation, as well as the risk of cardiovascular diseases during the COVID-19 pandemic [11]. Nagata et al. reported that among US adolescents, the odds of hypertension during the COVID-19 pandemic were approximately 2.00 times higher compared to the pre-pandemic period [12]. Schefelker et al., in their study, found that children with pre-existing dyslipidemia during the COVID-19 pandemic had worsening markers of the risk factors of atherosclerotic cardiovascular disease, including dyslipidemia, diabetes, and insulin resistance, compared to their markers from the previous year [13]. Eckert et al. observed significant increases in both systolic blood pressure (SBP) and diastolic blood pressure (DBP), as well as elevated levels of atherogenic lipids, among children and young adults with type 1 diabetes during the COVID-19 pandemic in Germany [14]. Additionally, other researchers have also shown that the COVID-19 pandemic had a negative effect on cardiovascular health factors among adults [15,16,17]. Although research studies have investigated changes in body weight and lifestyle among children and adolescents during the COVID-19 pandemic, there remains a scarcity of data on cardiovascular disease risk factors, particularly HBP, during this period. Furthermore, there is a lack of research investigating the relationship between changes in weight status and HBP among children and adolescents during the COVID-19 pandemic. Public health strategies must prioritize the prevention of HBP and obesity during childhood, as they are the main modifiable risk factors for cardiovascular disease. Addressing these factors early can lower the incidence of non-communicable diseases in adulthood and alleviate their long-term effects on health, including morbidity and mortality. 

Research studies conducted in Lithuania have indicated a notable prevalence of HBP among preschool children aged 3 to 7 years (21.4%) [18] and schoolchildren aged 7 to 18 years (27%) [19], as well as overweight and obesity rates of 16.4% and 5.8%, respectively, among children and adolescents aged 7–17 years [20]. Thus, in the present study, we aimed to investigate the associations between changes in weight status and HBP among Lithuanian schoolchildren during the COVID-19 pandemic.

## 2. Materials and Methods

### 2.1. Study Population

The cross-sectional baseline study was performed in Kaunas district, the second-largest district in Lithuania, from November 2019 to 15 March 2020. This study was completed before the announcement of the COVID-19 quarantine. Using a stratified two-stage cluster sampling design, 3757 participants were selected from the 1st through the 12th grade of all 29 participating schools, ranging in age from 7 to 18 years. Among 3757 schoolchildren, 47 were excluded due to missing anthropometric data (weight and height). The detailed description of the methods used in this cross-sectional study was provided earlier [19,20]. The follow-up survey was conducted from November 2021 to April 2022 (at the end of the COVID-19 quarantine and during the time of the national emergency related to COVID-19), maintaining identical methodology and procedures to those of the baseline study. The mean duration and standard deviation (SD) of the follow-up of the subjects were 24.96 ± 2.13 months. The second period from November 2021 to April 2022 was extended by one month due to several factors. The school’s working policies and the children’s holidays played a significant role in this decision. To accommodate these breaks and ensure minimal disruption to the data collection process, it was necessary to prolong the period. This extension was crucial to gather all the required data comprehensively, allowing for a more accurate and complete analysis. The participants were 7 to 18 years old at baseline and 8 to 18 years old at follow-up. Among the 3757 individuals involved in the baseline study, 2430 subjects participated in the follow-up study (the response rate was 65%). Thus, we analysed the data of 2430 subjects aged 8 to 18 who participated in both the baseline and follow-up studies. 

The research received approval from the Kaunas Regional Biomedical Research Ethics Committee at the Lithuanian University of Health Sciences on 10 June 2019, protocol No. BE–2–42. Following the principles and standards outlined in the Declaration of Helsinki, written informed consent was obtained from the study participants as well as from the parent or guardian of each participant. Additionally, the study objectives were thoroughly explained to all the parties involved.

### 2.2. Blood Pressure Measurements

A trained team collected blood pressure and anthropometric measurements following a standardized protocol. In the morning at the school, a physician not wearing a white coat performed blood pressure measurements for the participants. The subjects were instructed to abstain from physical activity and to avoid consuming beverages containing caffeine, including energy drinks, coffee, black tea, and green tea, prior to undergoing measurements on the day of the study. Three blood pressure measurements were taken using an automatic blood pressure (BP) monitor (OMRON M6; OMRON HEALTHCARE CO., LTD, Kyoto, Japan), with 5-min intervals of rest while seated, following a 10-min period of seated rest. The average of these three measurements was used in the analysis. Mean arterial pressure (MAP) was calculated as the following: DBP + 0.412 × (SBP − DBP) [21]. Pulse pressure (PP) was determined by subtracting DBP from SBP. According to the 2017 AAP Clinical Practice Guideline for Screening and Management of High Blood Pressure in Children and Adolescents [22,23], for children under 13 years, normal BP was defined as below the 90th percentile for age, sex, and height, elevated BP was above the 90th and less than the 95th percentile for age, sex, and height, and hypertension was at or above the 95th percentile for age, sex, and height. In adolescents aged 13 years and older, normal BP was described as below 120/80 mm Hg, elevated BP ranged from 120/<80 to 129/<80 mm Hg, and hypertension was at or above 130/≥80 mm Hg. HBP for children and adolescents was defined as either elevated BP or hypertension. 

### 2.3. Anthropometric Measurements

After measuring BP, anthropometric measurements were conducted. The subjects’ height, measured barefoot, was carefully recorded to the nearest 0.1 cm using a portable stadiometer (HM-250P Leicester Height Measure, Marsden, UK). The weight of the participants, taken while they wore light attire and were barefoot, was meticulously measured to the closest 0.1 kg employing an OMRON BF511 body composition monitor. BMI was calculated by dividing body weight in kilograms (kg) by height in meters squared (m^2^). Overweight and obesity were determined using BMI cut-off points specific to age and sex, as suggested by the International Obesity Task Force (IOTF) [24]. The participants were divided into groups based on their body weight status at baseline and follow-up assessments: participants with normal body weight at both baseline and follow-up, participants who had overweight/obesity at baseline but had normal body weight at follow-up, participants who had normal weight at baseline but had overweight/obesity at follow-up, and participants who had overweight/obesity at both surveys. Detailed numbers and percentages of changes in the participants’ weight status from baseline to follow-up according to sex are presented in Table A1. Tri-ponderal mass index (TMI) was determined by dividing the weight in kilograms by the height in meters cubed (kg/m^3^). Waist circumference (WC) was measured at a level midway between the lower rib margin and the iliac crest, while hip circumference (HC) was measured at the widest point around the buttocks, both recorded to the nearest 0.5 cm with a flexible tape. Abdominal obesity was characterized as waist circumference (WC) at or above the 90th percentile, based on age- and sex-specific WC percentile cut-offs recommended by the Third National Health and Nutrition Examination Survey (NHANES III) [25]. Waist-to-height ratio (WHtR) was calculated as WC (cm) divided by height (cm), while waist-to-hip ratio (WHR) was calculated as WC (cm) divided by HC (cm). Neck circumference (NC) was measured at the thyroid cartilage level using a flexible tape measure, with subjects standing upright, head erect, and eyes facing forward, to the nearest 0.1 cm. The mid-upper arm circumference (MUAC) was assessed at the midpoint between the olecranon and acromion processes, with an accuracy of 0.1 cm. The seated participant’s dominant wrist circumference (WrC) was measured with a tape measure over the Lister tubercle of the distal radius and across the distal ulna, rounded to the nearest 0.1 cm [26]. The body roundness index (BRI) was calculated using the following formula [27]:$$BRI=364.2-365.5\times \sqrt{1-\left(\frac{{\left(WC/\left(2\pi \right)\right)}^{2}}{{\left(0.5\times height\right)}^{2}}\right)}$$

### 2.4. Statistical Analysis 

Categorical variables are expressed as numbers and percentages and were compared using the chi-square test and the z-test. The Kolmogorov–Smirnov test was used to assess the normality of the distribution of continuous variables. Normally distributed continuous variables were presented as means and standard deviations (SD), and group comparisons were conducted using the t-test. Univariate and multivariate logistic regression analyses were conducted to evaluate the associations between changes in different body weight status and HBP. Crude and adjusted odds ratios (OR and aOR, respectively) with 95% confidence intervals (CI) were calculated; in multivariate analyses, ORs were adjusted for age separately for boys and girls, whereas in the combined analysis for both sexes, ORs were adjusted for both age and sex. The analysis of the data was carried out using IBM SPSS for Windows, version 27.0 (IBM, Armonk, NY, USA); *p*-values less than 0.05 were considered to be statistically significant.

## 3. Results

Baseline (before the COVID-19 pandemic) and follow-up (during the COVID-19 pandemic) characteristics of the study subjects are shown in Table 1. The analysed sample consisted of 2430 subjects, of whom 52.5% (*n* = 1275) were boys, and 47.5% (*n* = 1155) were girls. The mean age of the subjects was 10.40 ± 2.50 years at baseline and 12.15 ± 2.42 years at follow-up. In both baseline and follow-up studies, boys were significantly taller and heavier, and they had significantly higher mean values of BMI, MUAC, NC, WC, WrC, BRI, WHtR, and WHR. Additionally, boys had significantly higher mean SBP and PP, along with significantly lower mean DBP compared to girls. Furthermore, there were no statistically significant differences observed in mean age, TMI, or MAP between the two groups. While boys exhibited a significantly higher mean HC than girls during the baseline survey, no differences between the groups were found in the follow-up study. In the baseline study (before the pandemic), the overall prevalence of HBP was 23.7% (25.7% for boys and 21.5% for girls), while in the follow-up survey during the COVID-19 pandemic, the prevalence of HBP increased to 27.4% (29.2% for boys and 25.4% for girls). Thus, HBP was more prevalent among boys than among girls both at baseline and at follow-up. According to the IOTF criteria, the prevalence of overweight and obesity was 17.1% and 5.9% at baseline (for boys: 18.8% and 7.0%; for girls: 15.2% and 4.8%), respectively, and 20.1% and 8.2% at follow-up (for boys: 22.2% and 10.0%; for girls: 17.7% and 6.3%), respectively. The prevalence of abdominal obesity (WC ≥ 90th percentile) was 5.6% (6.7% for boys and 4.3% for girls) at baseline and increased to 6.8% (8.6% for boys and 4.8% for girls) at follow-up. From baseline to the follow-up period, 67.7% of the subjects (63.8% of boys and 72.0% of girls) remained at a normal weight status, while 4% (3.9% of boys and 4.0% of girls) transitioned from overweight/obesity to a normal weight, 9.2% (10.4% of boys and 8.0% of girls) transitioned from normal weight to overweight/obesity, and 19.1% (21.9% of boys and 16.0% of girls) remained with overweight/obesity.

Overweight/obesity and abdominal obesity, observed both at baseline before the COVID-19 pandemic and during the follow-up period amid the COVID-19 pandemic, were more common among participants with HBP compared to those with normal BP (Table 2). Additionally, the transition from normal weight at baseline to overweight/obesity at follow-up, as well as persistent overweight/obesity from baseline to follow-up, were more common among individuals with HBP compared to those who were normotensive. In the follow-up survey, participants aged 13 years or older were more likely to have HBP compared to younger participants aged less than 13 years (37.2% vs. 20.5%), whereas those younger than 13 years were more likely to have overweight/obesity compared to older participants (31.1% vs. 24.4%). In both baseline and follow-up surveys, subjects with HBP had significantly higher mean values of age, weight, height, BMI, TMI, HC, MUAC, NC, WC, WrC, WHtR, SBP, DBP, MAP, and PP, compared to normotensive subjects. In boys and girls separately, the mean age was higher in the HBP group than in the normotensive group, but in girls, no significant difference between these groups in the mean age was found during the follow-up study (Table A2). 

In the multivariate analysis, following adjustment for age, significant associations were observed among boys who initially had normal weight but gained weight and developed overweight/obesity during the follow-up period (aOR = 2.40, *p* < 0.001), yet no significant associations were observed in girls (Table 3). Similarly, subjects who maintained overweight/obesity from baseline to follow-up showed increased odds of adverse outcomes (aOR = 3.13 for boys, *p* < 0.001; aOR = 2.13 for girls, *p* < 0.001), compared to those who remained with normal weight throughout the study period. In multivariate models adjusted for age and sex, for both boys and girls combined, the corresponding aORs were 1.14 (*p* = 0.598), 1.95 (*p* < 0.001), and 2.58 (*p* < 0.001) for changes in different weight status. Higher aORs were observed among children and adolescents who transitioned from a normal weight status at baseline to overweight/obesity at follow-up, while the highest aORs were observed among participants with persistent overweight/obesity. Children and adolescents who had overweight/obesity at baseline but achieved a normal weight status during follow-up did not have significantly increased odds of HBP.

## 4. Discussion

To our knowledge, this is the first cohort study in Lithuania, as well as in the Baltic countries, that investigated the associations between changes in weight status and HBP in children and adolescents over a period of approximately 2 years (a follow-up duration of 24.96 ± 2.13 months), which encompassed the COVID-19 pandemic period. In this study, we analysed the prevalence of overweight, obesity, and HBP, identified changes in body mass index categories, and assessed their impact on HBP in Lithuanian schoolchildren during the pandemic period.

Childhood obesity is rising at alarming rates worldwide, with its prevalence increasing dramatically during the COVID-19 pandemic, as confirmed by data from various countries linking weight gain in children and adolescents during this period to COVID-19 quarantine measures [28]. Obesity has a negative impact on an individuals’ health and is associated with an increased risk of cardiovascular disease [29]. In the current study, we observed an increase in the prevalence of HBP by 3.7%, overweight by 3.0%, obesity by 2.3%, and abdominal obesity by 1.2% among children and adolescents from baseline before the COVID-19 pandemic to follow-up during the COVID-19 pandemic. Additionally, we observed a rise in the prevalence of HBP and overweight/obesity among both boys and girls during this period, with boys showing notably higher prevalence rates of HBP and overweight/obesity compared to girls, which is partially consistent with findings from another cohort study where the proportion of both boys and girls with HBP increased, overweight/obesity being more prevalent in boys during the COVID-19 quarantine [30]. The analysis of the data from the Korea National Health and Nutrition Examination Survey 2018–2020 also identified an increase in the prevalence of hypertension among Korean children and adolescents during the COVID-19 outbreak, with rates increasing from 7.1% to 12.5%, especially noticeable among boys [31]. In a study by Maltoni et al., which included adolescents with obesity, SBP and DBP decreased without differences between boys and girls during the lockdown period, while weight, BMI, and WC increased in the whole sample; however, boys gained significantly more weight and reported spending more than twice the time with sedentary behaviours compared to girls [32]. 

The findings of epidemiological studies [33,34,35,36,37] resonate with our own observations, confirming the increase in overweight and obesity rates among children during the COVID-19 pandemic. Weaver et al. reported that children from a southeastern U.S. state were by 1.80 times more prone to overweight or obesity during the COVID-19 pandemic, compared to the years before the pandemic [38]. Jenssen et al. found a significant increase in the overall prevalence of obesity (from 13.7% to 15.4%) among 2- to 17-year-old patients within one year since the onset of the COVID-19 pandemic, with particularly higher increases noted in the age group of 5 to 9 years [39]. Woolford et al. observed a significant absolute increase in overweight or obesity among children and adolescents of various age groups during the pandemic, with the highest increase observed among 5- to 11-year-olds [40]. Azrak et al., in a follow-up study involving 6- to 9-year-old children in Argentina, reported an increase in the proportion of subjects with overweight/obesity, especially among boys, during the long-term pandemic lockdown [41]. Moliterno et al. found that among children in Vienna, Austria, the prevalence of obesity was high from 2017 to 2023; however, the highest peak was observed during the COVID-19 pandemic period in 2020 [37]. In a school-based survey conducted in China by Hu et al., a 2.4% increase in the prevalence of obesity (from 10.4% in 2017 to 12.8% during the COVID-19 pandemic in 2020) was observed, with significantly higher rates among boys [42]. A representative survey in Germany revealed that, based on parental responses, one in six children and adolescents (16%), particularly those from families with lower household income, those who were with existing overweight before the COVID-19 pandemic, and children aged 10–12 years, had increased weight gain—either slight or substantial—during the two years of the COVID-19 pandemic [43]. In a systematic review and meta-analysis, researchers observed a statistically significant increase in weight (mean difference, 1.65 kg) and a 2% rise in the prevalence of obesity among children during the COVID-19 pandemic [44]. 

In the present study, 9.2% of the participants transitioned from normal weight at baseline before the pandemic to overweight or obesity at follow-up during the COVID-19 pandemic, although other studies have reported higher percentages. For instance, Shalitin et al. found that among children and adolescents in Israel, 11.2% of those with normal weight before the pandemic period had overweight or obesity during the COVID-19 pandemic [34]. In a cohort study of Chinese school-aged children, 28.1% of those with a normal BMI shifted to overweight or obesity, and 42.4% of initially overweight children transitioned to obesity during the five-month quarantine period [30]. Over the period of the COVID-19 pandemic, two primary factors contributed to the potential increase in childhood obesity: decreased physical activity due to school closures and limited access or reduced time for physical education classes, sports, and outdoor play; and increased sedentary lifestyles resulting from distance learning and extended screen time. Additionally, unhealthy eating habits, social isolation, heightened stress, and anxiety also played significant roles [45,46]. Moreover, Piątkowska-Chmiel et al. highlighted the association between chronic stress related to the pandemic and the elevated risk of overweight and obesity in children and adolescents [47].

In the current study, we found that children and adolescents who transitioned from normal weight at baseline to overweight/obesity at follow-up, as well as those with persistent obesity from baseline to follow-up, exhibited significantly increased odds of HBP compared to subjects with persistent normal weight over the entire study period. This trend was evident in both our cohort study conducted during the pandemic and studies conducted before the pandemic [6,48]. Additionally, our study conducted during the COVID-19 pandemic identified the highest aORs among subjects with persistent overweight or obesity, aligning with findings from other studies conducted before the pandemic [6,7,48,49]. However, in another cohort study, adolescents who transitioned from the overweight or obesity category to normal weight presented the highest risk of hypertension compared to adolescents who remained in the normal BMI category at the age of both 13 and 17 years [50]. Research studies conducted before the COVID-19 pandemic found that children who transitioned from overweight or obesity at baseline to normal weight at follow-up had a reduced risk of hypertension [6] or HBP [48] or did not exhibit any significant risk of hypertension [49,50]. The results of our study showed that children and adolescents who had overweight/obesity at baseline before the COVID-19 pandemic but achieved normal weight at follow-up during the COVID-19 pandemic did not have significantly increased odds of developing HBP. In the National Longitudinal Study of Adolescent Health, researchers also found that losing weight in adulthood among those who had overweight/obesity in adolescence was not associated with increased odds of prehypertension and hypertension, except for black men [51]. A longitudinal cohort study by Fan et al. showed that individuals with a childhood BMI at or above the 75th percentile and adult BMI below the 75th percentile did not have greater cardiometabolic risks in adulthood compared to those with a BMI below the 75th percentile in both childhood and adulthood [52]. The meta-regression analysis, which involved studies with children and adolescents aged 4–19 years participating in lifestyle interventions, indicated that decreases in mean BMI-SDS greater than 0.7, 1, or 1.2 were likely to result in reductions in triglycerides, SBP, and low-density lipoprotein cholesterol, respectively [53]. However, during the follow-up period of the present research amid the COVID-19 pandemic, the participants were not involved in any lifestyle interventions. Since the COVID-19 pandemic began, the number of youths, including those under 18, at risk of cardiovascular disease has significantly increased due to pandemic-related lifestyle changes, resulting in unintended consequences like excessive weight gain, dyslipidemia, insulin resistance, and diabetes [54]. It is logical to assume that more children and adolescents experienced weight gain during quarantine because if they continued to eat as much as they did before but had significantly reduced physical activity due to lockdown restrictions, weight gain would naturally follow. However, this was not the case for everyone. There are numerous other components that could have contributed to this outcome, such as food quality, increased stress levels, disrupted sleeping conditions, and the heightened use of electronic devices. Since we did not investigate these additional factors, we cannot definitively determine the exact reasons for the weight gain. The interplay of these various influences means that multiple factors could be involved, making it difficult to pinpoint a single cause. Undoubtedly and consistently, public health interventions during childhood and adolescence are essential in the prevention of HBP and overweight/obesity in later life. Encouraging healthy lifestyle, healthy nutrition, and physical activity earlier in life can help to avoid excessive weight gain and to prevent HBP and obesity-related cardiometabolic complications. It is essential to continue research and to analyse the long-term impact of the COVID-19 pandemic on children’s health, and to observe the trends in HBP, overweight, and obesity among children and adolescents after the pandemic, as society returns to normal pre-pandemic life. Early identification of individuals at a higher cardiometabolic risk is also crucial for preventing later-life metabolic and cardiovascular diseases, thereby lessening their impact on individuals, society, and healthcare systems.

The strength of our research was a wide age range of Lithuanian schoolchildren aged 8–18 years. This cohort study aimed to evaluate the interplay between changes in body weight status and HBP among Lithuanian schoolchildren during the COVID-19 pandemic. However, it is important to recognize that our study also has some limitations. Firstly, our study was confined to a single district. Additionally, we did not conduct assessments of blood or biochemical parameters, nor did we gather data on the pubertal status of the participants. The validation of HBP diagnosis relied solely on BP measurements taken on the day of the study, both at baseline and at follow-up, averaging three readings. To confirm a diagnosis of hypertension, BP measurements should be conducted on at least three separate occasions. Relying on BP readings from a single visit may lead to an overestimation of hypertension prevalence. However, in epidemiological studies, it is common practice to take two or three BP measurements during a single visit to improve accuracy and mitigate this potential overestimation. Moreover, differences in the age of the examined schoolchildren, sample sizes, and the investigation of different durations of the COVID-19 pandemic, as well as the anthropometric measurements and BP methodologies in schoolchildren have also made it difficult to make comparisons across studies. During the follow-up, logistical hurdles arose due to the epidemic prevention measures, preventing us from obtaining blood pressure and anthropometric measurements from all participants who took part in the initial study prior to the pandemic. Some individuals relocated, while others, along with their parents or guardians, declined participation in the follow-up. The lockdown measures further impeded our research progress by restricting direct engagement with students and access to school facilities.

## 5. Conclusions

Following nearly two years of restrictions, the prolonged impact of the pandemic on body weight status and BP among children and adolescents was substantial. The results showed that the prevalence of overweight, obesity, abdominal obesity, and HBP increased in Lithuanian children and adolescents during the COVID-19 pandemic. The study suggests that transitioning from normal weight to overweight/obesity, especially persistent overweight/obesity, are linked to higher odds of HBP during the pandemic. Subjects who transitioned from overweight/obesity at baseline to normal weight during follow-up did not show significantly increased odds of developing HBP. Further large-scale, multi-ethnic follow-up studies involving paediatric populations are necessary to ascertain whether the adverse consequences of the COVID-19 pandemic will change and decrease.

## Figures and Tables

**Table 1 nutrients-16-03256-t001:** Demographic, anthropometric, and blood pressure (BP) characteristics of the participants according to sex.

Variables	Total (*n* = 2430)	Boys (*n* = 1275)	Girls (*n* = 1155)	*p* ^#^
**Characteristics in baseline study**				
BMI categories:				0.002
Normal weight	1870 (77.0)	946 (74.2)	924 (80.0) *	
Overweight	416 (17.1)	240 (18.8)	176 (15.2) *	
Obesity	144 (5.9)	89 (7.0)	55 (4.8) *	
BMI categories:				0.001
Normal weight	1870 (77.0)	946 (74.2)	924 (80.0) *	
Overweight/obesity	560 (23.0)	329 (25.8)	231 (20.0) *	
WC percentile categories:				0.010
<90th	2294 (94.4)	1189 (93.3)	1105 (95.7) *	
≥90th	136 (5.6)	86 (6.7)	50 (4.3) *	
BP categories:				0.014
NBP	1854 (76.3)	947 (74.3)	907 (78.5) *	
HBP	576 (23.7)	328 (25.7)	248 (21.5) *	
Age (years)	10.40 ± 2.50	10.43 ± 2.48	10.36 ± 2.51	0.438
Weight (kg)	41.24 ± 14.96	42.35 ± 15.88	40.03 ± 13.78	0.003
Height (cm)	147.37 ± 16.00	148.45 ± 16.81	146.17 ± 14.96	0.011
BMI (kg/m^2^)	18.39 ± 3.61	18.57 ± 3.67	18.19 ± 3.53	0.007
HC (cm)	77.50 ± 10.92	78.08 ± 10.94	76.85 ± 10.86	0.008
MUAC (cm)	21.58 ± 3.13	21.88 ± 3.26	21.25 ± 2.95	<0.001
NC (cm)	28.79 ± 6.83	29.56 ± 8.99	27.94 ± 2.75	<0.001
WC (cm)	62.12 ± 9.70	63.80 ± 10.30	60.27 ± 8.62	<0.001
WrC (cm)	14.08 ± 1.48	14.37 ± 1.54	13.77 ± 1.35	<0.001
BRI	4.12 ± 0.96	4.19 ± 0.96	4.05 ± 0.97	<0.001
TMI (kg/m^3^)	12.49 ± 2.11	12.53 ± 2.13	12.45 ± 2.08	0.327
WHR	0.80 ± 0.07	0.82 ± 0.06	0.79 ± 0.07	<0.001
WHtR	0.42 ± 0.05	0.43 ± 0.05	0.41 ± 0.05	<0.001
SBP (mm Hg)	107.81 ± 12.51	108.70 ± 13.03	106.82 ± 11.85	0.002
DBP (mm Hg)	62.23 ± 8.24	61.67 ± 8.23	62.84 ± 8.22	<0.001
MAP (mm Hg)	81.01 ± 8.65	81.05 ± 8.75	80.96 ± 8.55	0.860
PP (mm Hg)	45.58 ± 11.05	47.03 ± 11.72	43.98 ± 10.04	<0.001
**Characteristics in follow-up study**				
BMI categories:				<0.001
Normal weight	1742 (71.7)	864 (67.8)	878 (76.0) *	
Overweight	489 (20.1)	284 (22.2)	205 (17.7) *	
Obesity	199 (8.2)	127 (10.0)	72 (6.3) *	
BMI categories:				<0.001
Normal weight	1742 (71.7)	864 (67.8)	878 (76.0) *	
Overweight/obesity	688 (28.3)	411 (32.2)	277 (24.0) *	
WC percentile categories:				<0.001
<90th	2265 (93.2)	1165 (91.4)	1100 (95.2) *	
≥90th	165 (6.8)	110 (8.6)	55 (4.8) *	
BP categories:				0.035
NBP	1765 (72.6)	903 (70.8)	862 (74.6) *	
HBP	665 (27.4)	372 (29.2)	293 (25.4) *	
Age (years)	12.15 ± 2.42	12.18 ± 2.39	12.12 ± 2.44	0.348
Weight (kg)	51.50 ± 15.92	53.41 ± 17.52	49.39 ± 13.63	<0.001
Height (cm)	158.63 ± 13.65	160.32 ± 15.20	156.77 ± 11.43	<0.001
BMI (kg/m^2^)	20.06 ± 4.04	20.31 ± 4.19	19.79 ± 3.84	0.011
HC (cm)	84.13 ± 11.03	84.38 ± 11.34	83.85 ± 10.68	0.701
MUAC (cm)	23.22 ± 3.08	23.63 ± 3.28	22.77 ± 2.78	<0.001
NC (cm)	29.63 ± 2.92	30.46 ± 3.10	28.70 ± 2.37	<0.001
WC (cm)	65.63 ± 10.94	68.18 ± 11.67	62.82 ± 9.30	<0.001
WrC (cm)	14.61 ± 1.40	14.99 ± 1.49	14.19 ± 1.16	<0.001
BRI	3.60 ± 0.84	3.70 ± 0.89	3.50 ± 0.77	<0.001
TMI (kg/m^3^)	12.66 ± 2.34	12.69 ± 2.41	12.63 ± 2.26	0.979
WHR	0.78 ± 0.07	0.81 ± 0.07	0.75 ± 0.07	<0.001
WHtR	0.41 ± 0.06	0.43 ± 0.06	0.40 ± 0.05	<0.001
SBP (mm Hg)	107.97 ± 13.39	109.67 ± 14.46	106.09 ± 11.83	<0.001
DBP (mm Hg)	69.59 ± 7.98	68.72 ± 8.12	70.56 ± 7.72	<0.001
MAP (mm Hg)	85.40 ± 8.83	85.59 ± 9.24	85.20 ± 8.36	0.664
PP (mm Hg)	38.38 ± 11.74	40.95 ± 12.78	35.53 ± 9.71	<0.001
Changes in weight status from baseline to follow-up:				<0.001
Normal weight ^A^ and normal weight ^B^	1646 (67.7)	814 (63.8)	832 (72.0) *	
Overweight/obesity ^A^ and normal weight ^B^	96 (4.0)	50 (3.9)	46 (4.0)	
Normal weight ^A^ and overweight/obesity ^B^	224 (9.2)	132 (10.4)	92 (8.0) *	
Overweight/obesity ^A^ and overweight/obesity ^B^	464 (19.1)	279 (21.9)	185 (16.0) *	

Values are numbers (percentages) and mean ± SD (standard deviation). ^#^ Boys versus girls. * *p* < 0.05 as compared to boys (z-test). BMI—body mass index, HC—hip circumference, MUAC—mid-upper arm circumference, NC—neck circumference, WC—waist circumference, WrC—wrist circumference, BRI—body roundness index, TMI—tri-ponderal mass index, WHR—waist–hip ratio, WHtR—waist-to-height ratio, SBP—systolic blood pressure, DBP—diastolic blood pressure, MAP—mean arterial pressure, PP—pulse pressure. ^A^—body weight status at baseline (before the COVID-19 pandemic), ^B^—body weight status at follow-up (during the COVID-19 pandemic).

**Table 2 nutrients-16-03256-t002:** Characteristics of the study participants according to BP level.

Variables	Normal BP	HBP	*p* ^#^
**Characteristics in baseline study**			
**All participants**			
BMI categories:			<0.001
Normal weight	1518 (81.9)	352 (61.1) *	
Overweight	262 (14.1)	154 (26.7) *	
Obesity	74 (4.0)	70 (12.2) *	
BMI categories:			<0.001
Normal weight	1518 (81.9)	352 (61.1) *	
Overweight/obesity	336 (18.1)	224 (38.9) *	
WC percentile categories:			<0.001
<90th	1789 (96.5)	505 (87.7) *	
≥90th	65 (3.5)	71 (12.3) *	
Age (years)	10.00 ± 2.31	11.70 ± 2.62	<0.001
Weight (kg)	37.98 ± 12.76	51.74 ± 16.63	<0.001
Height (cm)	144.56 ± 14.92	156.39 ± 16.02	<0.001
BMI (kg/m^2^)	17.68 ± 3.12	20.66 ± 4.10	<0.001
HC (cm)	75.04 ± 9.68	85.39 ± 10.95	<0.001
MUAC (cm)	20.90 ± 2.72	23.78 ± 3.35	<0.001
NC (cm)	28.21 ± 7.48	30.64 ± 3.51	<0.001
WC (cm)	60.07 ± 8.15	68.72 ± 11.22	<0.001
WrC (cm)	13.77 ± 1.32	15.09 ± 1.52	<0.001
BRI	4.15 ± 0.93	4.03 ± 1.06	0.001
TMI (kg/m^3^)	12.26 ± 1.92	13.26 ± 2.48	<0.001
WHR	0.80 ± 0.07	0.80 ± 0.07	0.895
WHtR	0.42 ± 0.05	0.44 ± 0.07	<0.001
SBP (mm Hg)	102.99 ± 8.87	123.31 ± 9.61	<0.001
DBP (mm Hg)	60.08 ± 6.94	69.14 ± 8.31	<0.001
MAP (mm Hg)	77.76 ± 6.43	91.46 ± 6.31	<0.001
PP (mm Hg)	42.91 ± 8.96	54.17 ± 12.67	<0.001
**Characteristics in follow-up study**			
**All participants**			
BMI categories:			<0.001
Normal weight	1341 (76.0)	401 (60.3) *	
Overweight	331 (18.8)	158 (23.8) *	
Obesity	93 (5.2)	106 (15.9) *	
BMI categories:			<0.001
Normal weight	1341 (76.0)	401 (60.3) *	
Overweight/obesity	424 (24.0)	264 (39.7) *	
WC percentile categories:			<0.001
<90th	1688 (95.6)	577 (86.8) *	
≥90th	77 (4.4)	88 (13.2) *	
Age (years)	11.81 ± 2.26	13.06 ± 2.58	<0.001
Weight (kg)	47.98 ± 13.36	60.85 ± 18.23	<0.001
Height (cm)	156.34 ± 12.77	164.72 ± 14.05	<0.001
BMI (kg/m^2^)	19.32 ± 3.51	22.04 ± 4.65	<0.001
HC (cm)	81.97 ± 9.96	89.87 ± 11.66	<0.001
MUAC (cm)	22.60 ± 2.70	24.89 ± 3.42	<0.001
NC (cm)	29.02 ± 2.49	31.25 ± 3.31	<0.001
WC (cm)	63.57 ± 9.25	71.12 ± 13.01	<0.001
WrC (cm)	14.34 ± 1.25	15.33 ± 1.52	<0.001
BRI	3.60 ± 0.81	3.63 ± 0.93	0.846
TMI (kg/m^3^)	12.38 ± 2.13	13.40 ± 2.68	<0.001
WHR	0.78 ± 0.07	0.79 ± 0.08	0.001
WHtR	0.41 ± 0.05	0.43 ± 0.07	<0.001
SBP (mm Hg)	102.91 ± 10.08	121.39 ± 11.73	<0.001
DBP (mm Hg)	66.88 ± 5.98	76.80 ± 8.17	<0.001
MAP mm Hg)	81.72 ± 6.52	95.17 ± 6.38	<0.001
PP (mm Hg)	36.03 ± 9.15	44.59 ± 15.11	0.001
Changes in weight status from baseline to follow-up:			<0.001
Normal weight ^A^ and normal weight ^B^	1271 (72.0)	375 (56.4) *	
Overweight/obesity ^A^ and normal weight ^B^	70 (4.0)	26 (3.9)	
Normal weight ^A^ and overweight/obesity ^B^	150 (8.5)	74 (11.1) *	
Overweight/obesity ^A^ and overweight/obesity ^B^	274 (15.5)	190 (28.6) *	

Values are numbers (percentages) and mean ± SD (standard deviation). The chi-square (χ^2^) test was used for categorical variables. ^#^ NBP versus HBP. * *p* <0.05 as compared to NBP group (z-test). BMI—body mass index, HC—hip circumference, MUAC—mid-upper arm circumference, NC—neck circumference, WC—waist circumference, WrC—wrist circumference, BRI—body roundness index, TMI—tri-ponderal mass index, WHR—waist–hip ratio, WHtR—waist-to-height ratio, SBP—systolic blood pressure, DBP—diastolic blood pressure, MAP—mean arterial pressure, PP—pulse pressure. ^A^—body weight status at baseline (before the COVID-19 pandemic), ^B^—body weight status at follow-up (during the COVID-19 pandemic).

**Table 3 nutrients-16-03256-t003:** Associations between changes in weight status from baseline to follow-up and HBP.

Variables	Boys		Girls		Total	
	OR (95% CI)	aOR^1^ (95% CI)	OR (95% CI)	aOR^1^ (95% CI)	OR (95% CI)	aOR^2^ (95% CI)
Changes in weight status from baseline to follow-up:						
Normal weight ^A^ and normal weight ^B^	1.00	1.00	1.00	1.00	1.00	1.00
Overweight/obesity ^A^ and normal weight ^B^	1.28 (0.68–2.42) ^NS^	0.93 (0.46–1.90) ^NS^	1.23 (0.63–2.43) ^NS^	1.23 (0.63–2.44) ^NS^	1.26 (0.79–2.00) ^NS^	1.14 (0.70–1.85) ^NS^
Normal weight ^A^ and overweight/obesity ^B^	1.82 (1.23–2.69)	2.40 (1.58–3.64)	1.45 (0.90–2.34) ^NS^	1.53 (0.95–2.48) ^NS^	1.67 (1.24–2.26)	1.95 (1.43–2.66)
Overweight/obesity ^A^ and overweight/obesity ^B^	2.52 (1.89–3.35)	3.13 (2.29–4.28)	2.08 (1.48–2.92)	2.13 (1.52–3.00)	2.35 (1.89–2.92)	2.58 (2.06–3.23)

OR—odds ratio; aOR^1^—adjusted odds ratio for age, aOR^2^—adjusted odds ratio for age, sex; CI—confidence interval. ^A^—body weight status at baseline (before the COVID-19 pandemic), ^B^—body weight status at follow-up (during the COVID-19 pandemic), ^NS^—non-significant.

## Data Availability

According to the Statute of the Lithuanian University of Health Sciences, the authors cannot share the data underlying this study. For inquires on the data, researchers should first contact the owner of the database, the Lithuanian University of Health Sciences. The data are not publicly available due to ethical issues.

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
