# Peer review of "Associations between Changes in Body Weight Status and High Blood Pressure among Lithuanian Children and Adolescents during the COVID-19 Pandemic: A Retrospective Cohort Study"

_nutrients, 2024, doi:10.3390/nu16193256_

Round 1

Reviewer 1 Report

Comments and Suggestions for Authors

Dear authors thank you for an interesting article.

Please make background methods results and conclusion in your abstract bold, write those words in capitals and let each of them them start at a new line because that is the way to do it. Also, that way, it’s much easier to read.

You selected 3757 participants, from 29 participating schools. Please explain if you included all the children from those schools and if not why children were excluded .

Your first measurement was from November 2019 to March 2020. as you know quarantine started on the 16th of March 2020 does this mean that this first part included two weeks of quarantine? Or do you mean to the 1st of March 2020?

Why was the second period from November 2021 to April 2022, in other words one month longer?

Why was a blood pressure (120/80) not included in the normal blood pressure? 

Why was a diastolic pressure of 80 or more labelled as hypertension?

The only way to properly measure if someone has hypertension or not is my measuring it three times daily for a week. so you need to add that measuring it only once doesn’t tell you if people have hypertension or not. You need to add that to your abstract and conclusion.

Why did boys have a significantly lower mean diastolic blood pressure if they were heavier et cetera cause usually slimmer girls have lower systolic and diastolic blood pressure than boys.

In your discussion, you’re repeating yourself too much.

Please add a logical explanation why more children and adolescents had weight gain during quarantine because if you continue to eat as much as you did before but you hardly have any physical activities because of the lockdown then it’s logical that many of them will put on weight. 

Please describe how many boys and how many girls are overweight obese, et cetera at your first measurement and at the second one.

Why didn’t you repeat your measurements in 2024 so then you could’ve seen if many youngsters would go back to their pre-lockdown BMI

You also stated that some children and adolescence went from being overweight to having normal weight but how many of them? And why did they manage to lose weight during the lockdown?

Please change the first sentence of your conclusion and remove the parts that it had a severe impact on individuals' health because you don’t provide any explanation for that nor is it a logical conclusion from your article.

Comments on the Quality of English Language

Minor language editing needed.

Author Response

Comment 1: Please make background methods results and conclusion in your abstract bold, write those words in capitals and let each of them them start at a new line because that is the way to do it. Also, that way, it’s much easier to read.

Response 1: Thank you for your comments. The abstract was prepared according to MDPI Nutrients requirements.

Comment 2: You selected 3757 participants, from 29 participating schools. Please explain if you included all the children from those schools and if not why children were excluded.

Response 2: Thank you for your question. A total of 3757 participants (from the 1st through the 12th grade of all 29 participating schools; ages 7–18 years) were selected using a stratified two-stage cluster sampling design. All the invited primary schools, basic schools, pre-gymnasiums, and gymnasiums agreed to participate in the survey. Of 3757 schoolchildren, 47 were excluded owing to missing anthropometric data. The study included only students whose parents or guardians and they themselves signed consents to participate in the study.

Comment 3: Your first measurement was from November 2019 to March 2020. as you know quarantine started on the 16th of March 2020 does this mean that this first part included two weeks of quarantine? Or do you mean to the 1st of March 2020?

Response 3: Thank you for comment. The cross-sectional baseline study (before the COVID-19 pandemic) was performed in from November 2019 to 15 March 2020. This study was completed before the announcement of the COVID-19 quarantine.

Comment 4: Why was the second period from November 2021 to April 2022, in other words one month longer?

Response 4: Thank you for your comment. The second period from November 2021 to April 2022 was extended by one month due to several factors. The school's working policies and the children's holidays played a significant role in this decision. To accommodate these breaks and ensure minimal disruption to the data collection process, it was necessary to prolong the period. This extension was crucial to gather all the required data comprehensively, allowing for a more accurate and complete analysis.

Comment 5: Why was a blood pressure (120/80) not included in the normal blood pressure?  Why was a diastolic pressure of 80 or more labelled as hypertension?

Response 5: Thank you for your comments. According to the 2017 the American Academy of Pediatrics CPG, normal BP is defined: <120/<80 mm Hg; and hypertension – as BP ≥130/≥80 mm Hg for adolescents ≥13 years old.

Comment 6: The only way to properly measure if someone has hypertension or not is my measuring it three times daily for a week. so you need to add that measuring it only once doesn’t tell you if people have hypertension or not. You need to add that to your abstract and conclusion.

Response 6: Thank you for your comment. We agree and add this information to page 12, line 398-402 “[To confirm a diagnosis of hypertension, BP measurements should be conducted on at least three separate occasions. Relying on BP readings from a single visit may lead to an overestimation of hypertension prevalence. However, in epidemiological studies, it is common practice to take two or three BP measurements during a single visit to improve accuracy and mitigate this potential overestimation.]”

Comment 7: Why did boys have a significantly lower mean diastolic blood pressure if they were heavier et cetera cause usually slimmer girls have lower systolic and diastolic blood pressure than boys.

Response 7: Boys tend to have lower diastolic blood pressure compared to girls during adolescence, possibly due to growth spurts and hormonal changes. The mean diastolic blood pressure of boys and girls according to BMI categories was not analyzed in the current manuscript.

Comment 8: In your discussion, you’re repeating yourself too much.

Response 8: Thank you for your comment. We revised the discussion paragraph and minimized the repetition.

Comment 9: Please add a logical explanation why more children and adolescents had weight gain during quarantine because if you continue to eat as much as you did before but you hardly have any physical activities because of the lockdown then it’s logical that many of them will put on weight. 

Response 9: Thank you for your comment. It's logical to assume that more children and adolescents experienced weight gain during quarantine because if they continued to eat as much as they did before but had significantly reduced physical activity due to lockdown restrictions, weight gain would naturally follow. However, this was not the case for everyone. There are numerous other components that could have contributed to this outcome, such as food quality, increased stress levels, disrupted sleeping conditions, and the heightened use of electronic devices. Since we did not investigate these additional factors, we cannot definitively determine the exact reasons for the weight gain. The interplay of these various influences means that multiple factors could be involved, making it difficult to pinpoint a single cause.

Comment 10: Please describe how many boys and how many girls are overweight obese, et cetera at your first measurement and at the second one.

Response 10: Thank you for your comment. The information was provided in detail in the first paragraph of the results.

Comment 11: Why didn’t you repeat your measurements in 2024 so then you could’ve seen if many youngsters would go back to their pre-lockdown BMI

Response 11: Thank you for your comment. In 2024, research on this topic is currently being continued among children and adolescents.

Comment 12: You also stated that some children and adolescence went from being overweight to having normal weight but how many of them? And why did they manage to lose weight during the lockdown?

Response 12: Thank you for your question. 3.7 percent of overweight children achieved a normal weight (Table A1). However, during the follow-up period of the present research amid the COVID-19 pandemic, the participants were not involved in any lifestyle interventions. As mentioned in the previous question, the weight loss observed in the participants may have been influenced by various components, that we did not investigate making it difficult to pinpoint a single cause.

Comment 13: Please change the first sentence of your conclusion and remove the parts that it had a severe impact on individuals' health because you don’t provide any explanation for that nor is it a logical conclusion from your article.

Response 13: Thank you for the comment. Corrections have been made in the revised version. Page 12, line 413-414 „[Following nearly two years of restrictions, the prolonged impact of the pandemic on body weight status and BP among children and adolescents was substantial.]”

Reviewer 2 Report

Comments and Suggestions for Authors

General comments:

The manuscript refers to a comparative study in a large Lithuanian child and youth population between the effects of the confinement decreed by the Lithuanian government on several parameters related to hypertension in this child and youth population. It is a bit striking that a work done between 2019 and 2021 is sent for publication in the latter part of the year 2024. It seems a very long time lag, but the work has been authorized by its corresponding ethics committee, and these procedures, together with the mandatory reports to the committee, can greatly delay the publication of the results.

In general, the manuscript draws conclusions that are already well known (the confinements decreed by the Covid pandemic have greatly worsened the weight situation and indicators of different metabolic diseases among the population). In this sense, the work only extends current knowledge, it does not create new knowledge. However, it is true that the confinements resulting from this pandemic have not been similar all over the world, so it is interesting to have epidemiological data from different parts of the world, as this may lead to evaluate which have been the most effective measures for future actions. It is true what the authors say that few data have been published from the countries in their geographical area, so I believe that this work may be useful.

From the scientific point of view, the work seems to be technically well done, and the defects found refer more to the formats used, than to the science used.

The grammar and spelling in English also seem correct, although I am not a native English speaker.

Specific comments:

Page 1, line 11: “High blood pressure was cited previously in line 9. Thus, it should be abbreviated as “HBP” the first time that appears in the text.

Page 1, line 34: Please define “NCD”.

Page 2, lines 63-89: Although the information presented here is interesting and relevant, I think this paragraph would be better placed in the discussion rather than in the introduction, as it makes the introduction too long and the reader loses the context of what the authors are presenting.

Page 3, line 111. What is SD? The same for BP in line 129.

Subheadings in materials and methods should be numbered and italicized according to the Nutrients format.

Page 4, line 187: “P” can be named as capital or ordinary characters but being uniform throughout the manuscript.

Page 4, line 195. All these abbreviations must be defined the first time that appears in the text.

The text of the Tables seems appears to be written in a larger font size than the rest of the text and larger than recommended in the instructions for authors. Also, if you reduce this font size, it is possible that the size of the tables will be more manageable and will not span several pages in Table 1.

Page 20: Please insert a space previously to “Disclaimer”.

Author Response

Comment 1: The manuscript refers to a comparative study in a large Lithuanian child and youth population between the effects of the confinement decreed by the Lithuanian government on several parameters related to hypertension in this child and youth population. It is a bit striking that a work done between 2019 and 2021 is sent for publication in the latter part of the year 2024. It seems a very long time lag, but the work has been authorized by its corresponding ethics committee, and these procedures, together with the mandatory reports to the committee, can greatly delay the publication of the results.

Response 1: Thank you for your insightful feedback. We agree that the time lag between the completion of the study and the submission for publication is indeed notable. However, as you correctly pointed out, the process was influenced by the necessary approvals from the corresponding ethics committee, along with the extensive reporting required. These procedural steps, while essential to ensuring the integrity and ethical standards of the research, did contribute to the delay in the publication of our findings. We appreciate your understanding of the complexities involved in such research processes.

Comment 2: In general, the manuscript draws conclusions that are already well known (the confinements decreed by the Covid pandemic have greatly worsened the weight situation and indicators of different metabolic diseases among the population). In this sense, the work only extends current knowledge, it does not create new knowledge. However, it is true that the confinements resulting from this pandemic have not been similar all over the world, so it is interesting to have epidemiological data from different parts of the world, as this may lead to evaluate which have been the most effective measures for future actions. It is true what the authors say that few data have been published from the countries in their geographical area, so I believe that this work may be useful.

Response 2: Thank you for your insightful feedback. While we recognize that our manuscript reinforces well-established findings about the impact of COVID-19 confinements on weight and metabolic health, our goal was to contribute specific epidemiological data from our region, which has been underrepresented in the literature. We agree that understanding these regional variations is important for evaluating the effectiveness of different public health measures. We appreciate your acknowledgment of the value this data brings to the broader discussion.

Comment 3: From the scientific point of view, the work seems to be technically well done, and the defects found refer more to the formats used, than to the science used. The grammar and spelling in English also seem correct, although I am not a native English speaker.

Response 3: Thank you. 

Comment 4: Page 1, line 11: “High blood pressure was cited previously in line 9. Thus, it should be abbreviated as “HBP” the first time that appears in the text.

Response 4: Thank you for your observation, we change this in page 1, line 10-12 “[High blood pressure (HBP), overweight, and obesity are common, growing public health problems worldwide. The aim of this study was to evaluate associations between changes in body weight status and HBP among Lithuanian children and adolescents during the COVID-19 pandemic.]”.

Comment 5: Page 1, line 34: Please define “NCD”.

Response 5: Thank you for your observation, we change this in page 1, line 35-36 “[According to the analysis conducted by the Non-Communicable Diseases (NCD) Risk Factor Collaboration (NCD-RisC), in 2022, 94.2 million boys and 65.1 million girls]”.

Comment 6: Page 2, lines 63-89: Although the information presented here is interesting and relevant, I think this paragraph would be better placed in the discussion rather than in the introduction, as it makes the introduction too long and the reader loses the context of what the authors are presenting.

Response 6: Thank you for your comment, although we do not agree, because according to Nutrients instructions we must highlight controversial and diverging hypotheses of other key publications, which we did in this paragraph. We believe it is at most importance to keep this paragraph in the introduction section.

Comment 7: Page 3, line 111. What is SD? The same for BP in line 129.

Response 7: Thank you for your observation, we explained this in page 3, line 112 and 131 “[standard deviation – SD; blood pressure – BP]”

Comment 8: Subheadings in materials and methods should be numbered and italicized according to the Nutrients format.

Response 8: Thank you for your observation, we agree, and we modified subheadings to emphasize this point.

Comment 9: Page 4, line 187: “P” can be named as capital or ordinary characters but being uniform throughout the manuscript.

Response 9: Thank you for your observation, we modified the “p” character to be uniform throughout the manuscript.

Comment 10: Page 4, line 195. All these abbreviations must be defined the first time that appears in the text.

Response 10: Thank you, although all those abbreviations are defined in the text, mostly in the materials and methods section.

Comment 11: The text of the Tables seems appears to be written in a larger font size than the rest of the text and larger than recommended in the instructions for authors. Also, if you reduce this font size, it is possible that the size of the tables will be more manageable and will not span several pages in Table 1.

Response 11: Thank you for your observation. Although we see why the text would seem bigger in the tables than the rest of the text, we assure you, the font size is the same and adheres to instructions for authors.

Comment 12: Page 20: Please insert a space previously to “Disclaimer”.

Response 12: Thank you, we modified it.

Round 2

Reviewer 1 Report

Comments and Suggestions for Authors

Dear authors, please address all my comments from the first round of peer review. Also please state in a letter how you have addressed them.

please also state what the response rate is.

Comments on the Quality of English Language

Minor language editing needed

Round 3

Reviewer 1 Report

Comments and Suggestions for Authors

Dear authors thank you for making the changes.

Author Response

Dear reviewer,

Thank you for your thorough review and valuable feedback. We greatly appreciate your insights, which have helped us improve the clarity and quality of our manuscript.